# Determinants of the Impact of Fatigue on the Health of Polish Nursing Students during the COVID-19 Pandemic

**DOI:** 10.3390/jcm11206034

**Published:** 2022-10-13

**Authors:** Ewa Kupcewicz, Kamila Rachubińska, Aleksandra Gaworska-Krzemińska, Anna Andruszkiewicz, Ewa Kawalec-Kajstura, Dorota Kozieł, Katarzyna Młynarska, Elżbieta Grochans

**Affiliations:** 1Department of Nursing, Collegium Medicum, University of Warmia and Mazury in Olsztyn, 10-719 Olsztyn, Poland; 2Department of Nursing, Pomeranian Medical University in Szczecin, 71-210 Szczecin, Poland; 3Institute of Nursing and Midwifery, Medical University of Gdansk, 80-227 Gdansk, Poland; 4Department of Basic Clinical Skills and Postgraduate Education for Nurses and Midwifes, Nicolaus Copernicus University in Torun, 85-821 Bydgoszcz, Poland; 5Department of Internal Medicine and Community Nursing, Institute of Nursing and Midwifery, Faculty of Health Sciences, Medical College, Jagiellonian University, 30-688 Krakow, Poland; 6Medical College, J. Kochanowski University in Kielce, 25-369 Kielce, Poland

**Keywords:** COVID-19, pandemic, fatigue, health

## Abstract

(1) Lockdown-related fatigue occurring during the COVID-19 pandemic is a complex problem that can be experienced in different social groups. The objective of the current study is to attempt to identify socio-demographic and lifestyle-related factors that determine the impact of fatigue on health in general as well as in physical, cognitive, and psychosocial terms and to determine whether, and to what extent, these were predictors of fatigue in nursing students during the COVID-19 pandemic. (2) The study was conducted by the diagnostic poll method between 20 March and 15 December 2021 among 894 nursing students at six Polish universities. To collect the data, a validated Modified Fatigue Impact Scale (MFIS) was used. (3) Students from the age group of ≤20 experienced a significantly greater impact of fatigue on health in general as well as in physical and cognitive terms. The study demonstrated a significant negative relationship between the year of study and the impact of fatigue on health in general terms (r = −0.12; *p* < 0.0001) and the analyzed health terms, on physical (r = −0.12; *p* < 0.0001), cognitive (r = −0.10; *p* < 0.002), and psychosocial (r = −0.07; *p* < 0.041). In predicting the impact of fatigue on health in general and physical terms, it was the variable related to a reduction in physical activity during the COVID-19 pandemic that had the greatest contribution, while for the cognitive and psychosocial functions, it was the number of meals consumed per day. (4) It is recognized that action is needed to reduce the impact of fatigue on student health by modifying the predictors related to student lifestyles.

## 1. Introduction

The COVID-19 pandemic, recognized by the World Health Organization (WHO) as the greatest pandemic in modern times, and the associated need for the introduction of drastic measures to limit the spread of the disease, had an enormous impact on the lives of people around the world and resulted in changes in almost all spheres of their daily functioning [1,2,3,4]. It particularly affected young people who, on the one hand, are the part of the population that is least exposed to the adverse health consequences of infection, while on the other hand, are a group of people most affected by the psychosocial effects of the pandemic. In order to cut the SARS-CoV-2 virus transmission pathways, the introduced restrictions involved, e.g., forced social distance, changes to the existing learning patterns, and the necessity to work remotely, or orders to stay at home, which resulted in limiting numerous forms of life activity [5,6,7,8,9].

Many researchers recognize the feeling of fatigue, which is a complex problem and may occur in different social groups, as one of the significant and frequently reported consequences of forced social isolation. The literature on the subject defines fatigue as a subjective feeling of energy loss as well as excessive exhaustion after performing habitual activities, which result from a change to the previous behavioral control mechanisms [10,11,12]. The responsible factors include, e.g., previously existing physical or mental health problems or negative lifestyle-related behaviors, including a poor diet, lack of sleep, or insufficient physical activity levels [10,13]. However, fatigue, referred to as “lockdown fatigue”, is defined as a state of energy loss and exhaustion resulting from the restrictions imposed due to the pandemic (especially the forced isolation), affecting all aspects of the biopsychosocial functioning of humans [14,15]. There are numerous reasons behind this state of affairs, one of them being the loss of security arising from the introduction of the COVID-19 pandemic state, associated with the inability to forecast and predict the future [6]. Another significant cause of lockdown-induced fatigue identified by researchers is the necessity to significantly modify the existing routine activities, not only in the professional or educational sphere but also in the realization of health-related behaviors such as physical activity, dietary habits, sleep and rest habits, and the extended time of the use of the Internet, i.e., the main tool for work and study, and often the only platform for interpersonal contacts [9,15,16,17,18,19]. The change in habits imposed on young people, and the resulting disruption of their previous functioning, has led to numerous adverse health consequences [6,10,20,21,22,23,24]. The deterioration of health in physical, mental, and behavioral terms appears to be the proven and most important consequence of lockdown-induced fatigue. It is manifested by numerous abnormalities of a psychosocial and somatic nature, including decreased interest in previously enjoyable activities, reduced motivation to take action, difficulty in controlling emotions, irritability, increased anxiety levels, depression, or excessive worrying [6,10,23]. The somatic effects of lockdown-induced fatigue, most frequently mentioned in the literature, include general weakness, sleep problems, headache, muscle pain, and back pain [16,25,26,27].

It is, therefore, reasonable to regard the fatigue induced due to the destructive effect of the COVID-19 pandemic on the functioning of young people as a serious problem. Indeed, not only does lockdown fatigue hinder the fulfilment of many social roles and expectations, but it also impairs the health of individuals, which may become a cause of decreased overall life satisfaction [6,16,26]. One of the ways to reduce the adverse impact of fatigue on health is to take actions aimed at modifying its predictors, especially those related to lifestyle, hence the attempt to identify them in the current study [28].

The restrictions imposed in an attempt to curb the worldwide spread of COVID-19 resulted in a rapid shift from the traditional, face-to-face teaching methods to the online teaching mode [29], thus contributing to increased learning difficulties among students [30]. Moreover, students had to cope with limitations in their daily lives, social relationships, spending free time and learning [31,32,33], which brought about significant changes in the daily routines of these young adults. These changes, in turn, contributed to increased feelings of loneliness, insecurity, isolation, and difficulty in maintaining contact with loved ones, peers, and academic staff [34]. The results of numerous studies have demonstrated that stress among nursing students was at a high level during the pandemic [35,36]. Other researchers demonstrated that students were very susceptible to mental health problems which actually escalated during the pandemic [37,38,39].

The objective of the study is to identify socio-demographic and lifestyle-related factors that determine the impact of fatigue on health in general as well as physical, cognitive, and psychosocial terms, and to determine whether and, if so, to what extent these were predictors of fatigue in nursing students during the COVID-19 pandemic.

In view of the presented scientific reports, it was hypothesized that:

Due to the restrictions imposed in relation to the SARS-CoV-2 virus transmission, students from the youngest age group (≤20) and the first-year nursing students experienced significantly greater fatigue impact on health in general, physical, cognitive, and psychosocial terms than students from older age groups, and more experienced ones.The situational factors arising from the restrictions imposed during the COVID-19 pandemic (e.g., the degree of physical activity restriction, time spent working on a computer, the degree of social contact restriction, and the number of meals consumed) are the predictors of the effect of fatigue on the health of nursing students in general, physical, cognitive, and psychosocial terms.

## 2. Materials and Methods

### 2.1. Settings and Design

The study was conducted in the second year of the COVID-19 pandemic, from 20 March to 15 December 2021, among nursing students at six Polish universities. Students numbering 894 participated in the study, including 143 (16.00%) from the Medical University of Gdańsk, 175 (19.57%) from the University of Warmia and Mazury in Olsztyn, 132 (14.77%) from the Jagiellonian University in Kraków, 171 (19.24%) from the Nicolaus Copernicus University in Toruń, Collegium Medicum in Bydgoszcz, 57 (6.38%) from the Jan Kochanowski University in Kielce, and 215 (24.05%) from the Pomeranian Medical University in Szczecin. The study was conducted after obtaining permission from the authorities at particular universities while respecting the sanitary regime rules. The criteria for inclusion in the study included students below 30 years old and giving informed consent to participate in the study. Individuals who did not consent to participate in the study were excluded. The respondents were informed of the objective of the study and the manner of filling in the survey questionnaires and had the opportunity to ask questions and receive comprehensive answers. The respondents were guaranteed full anonymity, and the collected data were analyzed collectively. A total of 975 questionnaire form sets were distributed. All of the respondents provided answers (975); however, after eliminating the incompletely filled-in responses, a total of 894 correctly filled-in sets (i.e., 91.69% of the total) were included in the study.

The questionnaire form completion time was approximately 15 min. The study presented here is part of a wider research project and satisfies the criteria for a cross-sectional study [24]. The project received a favorable opinion (No 3/2021) from the Senate Review Board and the Ethics Committee of the Olsztyn School of Higher Education in Olsztyn.

### 2.2. Participants

A total of 894 nursing students participated in the study, including 822 females (91.95%) and 72 males (8.05%). The mean age of the subjects was 20.73 years (SD = 1.81). The most numerous group comprised students aged 20 years and younger. A rather large group was that of first-year students (n = 397; 44.41%), while the second-year students accounted for 32.33% (n = 289), and the third-year students accounted for 23.27% (n = 208). The majority of the student subjects lived with their family or a relative. The duration of remote work was analyzed in three time intervals: ≤5 h, 6–9 h, and ≥10 h. The average number of hours worked remotely was 6.08 (SD = 3.19). Almost half of the respondents reported that they worked for ≤5 h per day, while 17.79% (n = 159) worked for as many as ≥10 h. The average number of meals consumed was 3.48 (SD = 0.87) per day. Every fourth (n = 21; 23.60%) subject stated that he/she had not reduced physical activity during the COVID-19 pandemic. Almost all respondents indicated that they enjoyed good or very good health. The reduction in social contact during the pandemic affected, to a significant degree, 40.27% of respondents (n = 360) (Table 1).

### 2.3. Research Instruments

The study was conducted by the diagnostic poll method, and the validated Modified Fatigue Impact Scale (MFIS) (modification by A. Gruszczak et al.) in a Polish language version was used for the collection of data [25].

For the description of the study group characteristics, a self-prepared questionnaire was used, which contained questions about the basic socio-demographic data and selected elements of the lifestyle, e.g., the number of hours spent working on the computer, the number of meals consumed, and the degree of restriction of social contact during the pandemic.

#### Modified Fatigue Impact Scale (MFIS)

The MFIS scale is comprised of 21 statements and is a tool for determining the impact of fatigue on health in physical terms and cognitive and psychosocial functions. The first part comprises nine questions concerning a subjective assessment of the impact of fatigue on physical functioning (Ph-MFIS). In the second part, the respondent provides responses to ten questions referring to the impact of fatigue on cognitive functions (C-MFIS). The third part is comprised of two questions determining psychosocial functions (Ps-MFIS). The respondent, when responding to questions, indicates the frequency of particular events related to fatigue over the past four weeks. The responses are assigned consecutive values ranging from 1 (never) to 5 points (almost always). The respondent can receive from 9 to 45 points for the assessment of physical functioning, from 10 to 50 for the assessment of cognitive functions, and from 2 to 10 points for the assessment of psychosocial functions. In the overall assessment using the MFIS questionnaire, the respondent can receive from 21 to 105 points. Since the scale has no specific cut-off point, the higher the score, the greater the intensity of the impact of fatigue on health. Validation analysis of the MFIS demonstrated satisfactory psychometric properties [40]. It was found that Cronbach’s alpha score for the Ph-MFIS subscale was 0.87; for the C-MFIS subscale, it was 0.85, while for the Ps-MFIS subscale, it was 0.69.

### 2.4. Statistical Analysis

The collected data were subjected to statistical analysis using a Polish version of STAT ISTICA 13 (TIBCO, Palo Alto, CA, USA). For a description of the analyzed variables, the mean value, standard deviation, median, minimum and maximum, and the 95% CI confidence interval for the mean value were used. An analysis of variance (ANOVA) was applied to indicate the probability with which the identified factors could be the reason for the differences between the observed group average values. The intergroup differences were determined by a post hoc (NIR) test. The correlations between selected socio-demographic and lifestyle factors and physical, cognitive, and psychosocial fatigue were analyzed using the r-Pearson correlation. The quantitative representation of correlations between multiple independent (explanatory) variables and the dependent variable is presented by means of multiple regression analysis. The interpretation of the correlation strength was based on the classification according to Guilford by adopting the following order: |r| = 0—no correlation, 0.0 < |r| ≤ 0.1—negligible correlation, 0.1 < |r| ≤ 0.3—poor correlation, 0.3 < |r| ≤ 0.5—average correlation, 0.5 < |r| ≤ 0.7—high correlation, 0.7 < |r| ≤ 0.9—very high correlation, 0.9 < |r| < 1.0—almost complete correlation, |r| = 1—complete correlation [41]. The statistical significance level was taken as *p* < 0.05.

## 3. Results

As can be seen from the data presented in Table 2, the overall index of the intensity of the impact of fatigue on health, in general terms, had a score of 57.67 points (SD = 15.02) on a scale ranging from 21 to 105. A subjective assessment of the impact of fatigue on physical functioning yielded an index of 24.55 points (SD = 7.02) on a scale ranging from 9 to 45. Statistical analyses revealed the average value of the result for the assessment of the impact of fatigue on cognitive functions as 27.79 points (SD = 8.70) on a scale ranging from 10 to 50, while the assessment of the impact of fatigue on psychosocial functions determined its value of 5.32 points (SD = 2.09) on a scale from 2 to 10 (Table 2).

Subsequently, the impact of fatigue on health in physical, cognitive and psychosocial terms was analyzed using the single factor variance analysis (ANOVA) in relation to the student’s age, the year of study, the number of hours spent working on the computer, the number of meals consumed, and the degree of reduction in physical activity and social contact during the COVID-19 pandemic. The NIR test was applied in parallel to identify possible intergroup differences.

### 3.1. Analysis of the Impact of Fatigue on Health in Physical Terms

The obtained statistical results show significant differences in the impact of fatigue on the physical health of students depending on age (F = 4.25; *p* < 0.01) and the year of study (F = 7.76; *p* < 0.0001). Students from the youngest age group (≤20) and those in their first year of study reported discomfort, due to deterioration in physical performance, significantly more frequently than second or third-year students. Moreover, differences were noted in the impact of fatigue on students’ physical health depending on the number of meals consumed (F = 4.25; *p* < 0.01). It was confirmed that the consumption of only one or two meals per day by the students was detrimental to health and significantly determined the level of the impact of fatigue on health in physical terms. As a result of the introduced restrictions related to the SARS-CoV-2 virus transmission, the respondents reduced different forms of physical activity, which significantly determines the level of fatigue in terms of physical health (F = 23.83; *p* < 0.0001). It was found that people who did not give up their physical activity were less prone to fatigue/weariness than people who even slightly reduced their physical activity. Detailed results are provided in Table 3.

### 3.2. Analysis of the Effect of Fatigue on Cognitive Functions

The analysis showed that the impact of fatigue on cognitive functions was determined by the students’ age (F = 5.18; *p* < 0.01) and the year of study (F = 4.94; *p* < 0.01).

Using a post hoc NIR test, it was found that students from the youngest age group (≤20) and those in their first year of nursing study, similar to physical fatigue, experienced a significantly greater impact of fatigue on cognitive functions, manifested, e.g., by impaired concentration, difficulty in making decisions, and lower motivation to perform thinking-related tasks. The number of hours spent working on the computer did not appear to be a factor that significantly differentiated the level of fatigue experienced by the students in cognitive terms. The results of a single factor variance analysis showed that significant differences occurred between the groups in the level of impact of fatigue on cognitive functions depending on the number of meals consumed by the students (F = 11.29; *p* < 0.0001). The highest level of the impact of fatigue on cognitive functions was noted for respondents who declared having only one or two meals a day. The reduction in physical activity during the COVID-19 pandemic also determined, to a significant extent, the degree of the impact of fatigue on health in cognitive terms (F = 4.04; *p* < 0.01). The students who did not give up various forms of physical activity experienced a significantly smaller impact of fatigue on health in cognitive terms compared to the individuals who reduced their physical activity during the pandemic to an average or significant extent. The results of significant difference tests, referring to the impact of fatigue on cognitive functions, are provided in Table 3.

### 3.3. Analysis of the Effect of Fatigue on Psychosocial Functions

The statistical analyses revealed that the degree of impact of fatigue on psychosocial functions varied (F = 3.34; *p* < 0.05) depending on the year of study of the students under study. First-year students significantly exhibited less motivation for social life and reduced their outdoor activities more than the second- or third-year students. During the pandemic, nursing students reduced physical activity, which has a preventive effect both in the context of lifestyle diseases and the functioning during infectious seasons, which significantly translated into an increase in the level of fatigue and its impact on health in psychosocial terms (F = 7.31; *p* < 0.0001). The number of meals consumed also significantly differentiated the level of the impact of fatigue on psychosocial functions in the group under study (F = 11.53; *p* < 0.0001). For individuals who consumed one or two meals per day, a greater fatigue impact on psychosocial functions was demonstrated than that for the individuals who consumed more meals. Due to online classes, the students spent long hours sitting in front of the computer. It was demonstrated that the number of hours working on the computer significantly differentiated (F = 3.76; *p* < 0.01) the level of impact of fatigue on the psychosocial health dimension. With an increase in the number of hours worked remotely, the level of fatigue related to the psychosocial aspect increased as well.

On the other hand, the students’ age and the reduction in social contact during the COVID-19 pandemic did not significantly differentiate the level of impact of fatigue on psychosocial functions. Detailed data are provided in Table 3.

### 3.4. The Degree of Intensity of the Correlation between Selected Socio-Demographic and Lifestyle-Related Factors and the Impact of Fatigue on Health in General, Physical, Cognitive, and Psychosocial Terms

In order to determine whether a statistically significant correlation existed between the analyzed variables, the r-Pearson correlation coefficient was calculated. The analysis of correlations showed a significant, negative relationship of a low strength between the year of study and the impact of fatigue on health in general terms (r = −0.12; *p* < 0.0001) and the analyzed dimensions of physical, (r = −0.12; *p* < 0.0001), cognitive (r = −0.10; *p* < 0.002), and psychosocial health (r = −0.07; *p* < 0.041). As can be seen, the students who had only just started their higher nursing education experienced the impact of fatigue on health in all its dimensions more intensely.

Further analyses showed that the number of hours spent working on the computer was positively correlated with the impact of fatigue on health in general terms. However, its strength was negligible yet still statistically significant (r = 0.09; *p* < 0.006). Having analyzed the relationship between the number of hours spent working on the computer and fatigue in particular health terms, only a positive relationship of low strength with an impact on health in physical terms was noted (r = 0.11; *p* < 0.002), while no relationship between the impact of fatigue and cognitive and psychosocial functions of the nursing students was noted.

The number of meals consumed also correlated with the impact of fatigue on health in general terms (r = −0.16; *p* < 0.0001), the physical health dimension (r = −0.13; *p* < 0.0000), and the cognitive (r = −0.14; *p* < 0.000) and psychosocial functions (r = −0.14; *p* < 0.000). All of the relationships noted took a negative direction and were of low strength but statistically significant. This means that if students eat rationally and consume an appropriate number of meals, the impact of fatigue on health diminishes, and vice versa.

Poor, yet significant, positive relationships were noted between the reduction in physical activity during the COVID-19 pandemic and the impact of fatigue on health in general terms (r = 0.19; *p* < 0.000), in physical terms (r = 0.26; *p* < 0.000), and cognitive functions (r = 0.09; *p* < 0.000). Further statistical analyses showed no relationship between the reduction in physical activity and the impact of fatigue on the nursing students’ psychosocial functions. The results of the presented analyses are provided in Figure 1.

### 3.5. Predictors of the Impact of Fatigue on Health in General, Physical, Cognitive, and Psychosocial Terms

Subsequent statistical analyses focused on the predictors of the impact of fatigue on health in general, physical, cognitive, and psychosocial terms. Several explanatory variables (i.e., the year of study, the number of meals consumed, reduction in physical activity and social contact during the COVID-19 pandemic) were introduced into the multiple regression model to predict the value of the dependent variable (the impact of fatigue on health in general terms-MFIS, and in physical (Ph-MFIS), cognitive (C-MFIS), and psychosocial (Ps-MFIS) terms.

Regression analysis demonstrated that the dependent variable concerning the impact of fatigue on health in general terms among nursing students was linked by a relationship with four explanatory variables that explained 8% of the variation in results (R^2^ = 0.08). The variable related to a reduction in physical activity during the pandemic, which explained 4% of the variation of the impact of fatigue on health in general terms, was ranked first. Each successive explanatory variable introduced into the model increased the determination coefficient (R^2^) value to a small extent (the number of meals consumed—2%; the year of study—1%; reduced social contact during the pandemic—1%) (Table 4). 

The prediction of the impact of fatigue on physical health involved four variables which explained 10% of the variation in results (R^2^ = 0.10). The largest proportion was attributed to the variable related to a reduction in physical activity during the pandemic, which explained 7% of the variation of results. The three remaining variables, i.e., the number of meals consumed, the year of study, and reduced social contact during the pandemic, represented a small proportion (a total of 3%) in the prediction of the impact of fatigue on the nursing students’ physical health.

Further considerations proved that the dependent variable concerning the impact of fatigue on health in cognitive terms among the nursing students was linked by a relationship with three explanatory variables that explained only 4% of the variation in results (R^2^ = 0.04). The number of meals consumed (2%) was ranked first and was followed by the reduction in social contact during the pandemic (1%) and the year of study (1%), respectively.

Similarly, in relation to the prediction of the impact of fatigue on health in psychosocial terms, two variables were involved, which explained only 3% of the variation (R^2^ = 0.03). The number of meals consumed was ranked first and explained 2% of the variation, while the reduction in physical activity during the pandemic explained 1% of the variation of results.

In summary, it should be indicated that although the obtained values of determination coefficients (R^2^) are low, the regression models under consideration exhibit statistical significance.

## 4. Discussion

Fatigue is currently a problem that affects people of increasingly younger age with increasing intensity, and its consequences disturb an individual’s functioning at different levels, both individually and in family and social terms [11,42,43,44]. Moreover, the longer it lasts and the greater its intensity, the more it brings disorganization to the performance of daily activities and duties inherent in various social roles. In addition, certain authors suggest that fatigue may not so much be a consequence of an objectively perceived burden of work or study but of a subjective sense of overload of responsibilities [45,46,47,48,49].

Numerous studies indicate lockdown-related fatigue as one of the most frequently reported consequences of the COVID-19 pandemic among young people [6,12,29]. At the same time, a number of its possible psychosocial and somatic symptoms have been indicated [6,10,16,23,24,25,26,50,51,52,53].

The nursing students, when participating in clinical classes, were particularly affected, both mentally and emotionally, by the need to care for patients who could potentially be infected with the SARS-CoV-2 virus. As demonstrated by the study, nurses need to be prepared for multi-dimensional activity in the area of communication with the patient and in the psychological care and physical care under difficult conditions of infection hazards [54,55,56,57,58].

Therefore, the above justifies the need to search for determinants of the impact of fatigue on health, especially those that can be modified.

The current study attempted to search for socio-demographic and lifestyle-related factors that determine the impact of fatigue on health in general as well as physical, cognitive, and psychosocial terms, and to determine whether, and to what extent, these are predictors of fatigue in a group of Polish nursing students during the COVID-19 pandemic.

In the authors’ own study, the overall index of the intensity of the impact of fatigue on health according to the Modified Fatigue Impact Scale (MFIS), in general terms, amounted to 57.67 points, which suggests a moderate level. On the other hand, for physical functioning, an index with a value of 24.55 points was obtained. For the cognitive aspect, the index was 27.79 points, and for the psychosocial aspect it was 5.32 points. The results of the current study are consistent with those of previous multi-center studies in which the fatigue severity, in general terms, in the group of Polish students achieved a score of 55.53 points. In addition, the authors demonstrated, for nursing students from three European countries, that the fatigue severity was significantly greater among Polish and Slovakian students than in the Spanish group [59].

Other studies, which assessed the fatigue levels using the Lockdown Fatigue Scale, also observed moderate fatigue levels in the group of students from various colleges and universities in West Samar, Philippines [6].

Moreover, a study by Sfeir et al., which assessed work fatigue using the 3D-WFI (3D-Work Fatigue Inventory) questionnaire, demonstrated that out of 401 medical students and doctors under study, 66.1%, 64.8%, and 65.1% respondents showed medium or high levels of emotional, mental, and physical work fatigue, respectively [60].

The current study also demonstrated significant differences in the impact of fatigue on students’ health in physical, cognitive, and psychosocial terms, depending on their age, the year of study, the number of meals consumed, and physical activity. Students from the youngest age group and those in their first year of study reported discomfort due to deterioration in physical performance and experienced a significantly greater impact of fatigue on cognitive functions, e.g., manifested by impaired concentration, difficulty in making decisions, and lower motivation to perform thinking-related tasks significantly more often than students in their second or third year of study. In addition, the first-year students significantly more often demonstrated less motivation for social life and reduced outdoor activities compared to the second- or third-year students. Therefore, the current study confirms the reports by Labrague et al. [6], which showed significantly lower levels of fatigue among graduates as compared to students at lower levels of education. The above relationship can be explained by the acquisition of adaptive behaviors or positive skills of coping with stressful situations by the students during their education [6]. In contrast, different results were obtained in a study by Liu et al. The results of their multi-center study involving 1070 nursing students demonstrated that the first-year students experienced fatigue significantly less often than the second-, third-, and fourth-year students. This unexpected result of their own study is explained by the authors by the fact of the greater burden associated with studying, taking more measures aimed at combating the pandemic, and the related greater responsibility of nursing students at a higher educational level, as compared to the first-year students [10].

Interestingly, the authors’ own study also demonstrated the destructive impact of reduced physical activity on health in all the areas under consideration. It was shown that people who did not give up their physical activity were less prone to fatigue than people who even slightly reduced their physical activity. Meanwhile, the positive effect of lifestyle-related health behaviors was also found, including physical activity and good nutrition, not only in the context of lifestyle disease prevention [61,62] but also in relation to maintaining and building personal immunity protecting against the SARS-CoV-2 virus infection [63]. Moderate daily physical activity and an appropriate diet ensuring the provision of all essential nutrients should be the basis for maintaining good health [63], which, according to the authors, may also translate into lower lockdown-related fatigue [64].

Currently, an increasing number of authors, when searching for predictors of lockdown-related fatigue, indicate a variety of variables, including socio-demographic variables [6,15,65] or an individual’s resources such as the skills of coping and personal immunity [15,60]. The above studies identified gender as an important predictor of fatigue. This is because females experienced higher levels of fatigue as compared to males, which might have been due to gender differences in expressing feelings and emotions, and to their expressing physical discomfort [6]. However, the study by Liu et al. noted that male students reported fatigue significantly more often than females [10]. This difference may be due to a different role assigned to males in traditional Chinese culture than that assigned to females and to the fact that considerably greater responsibility for challenges related to crisis situations, with the COVID-19 pandemic being undoubtedly one of them, is expected from males. The factor related to the disparity between social expectations in relation to males’ personal resources may therefore explain the fact that males experience fatigue more frequently than females. This observation is confirmed by the results of other studies, which confirmed that lower fatigue levels were linked to considerably higher levels of personal immunity and coping skills [15]. Increased physical activity is associated with the improvement of health in physical and mental terms and better coping with pandemic-induced stress [66]. Some authors also indicate additional activity as a factor that may serve as a protective role against fatigue. Kutyło et al. [45] demonstrated in a study that the fatigue levels in the group of working students were significantly lower than that in the group of non-working students. The authors suggest that working students may satisfy a proportion of the needs that are important to them in the work environment, and due to the need to effectively combine studying with work, they must be characterized by organizational skills, which help them develop more effective methods for managing time and responsibilities, which may protect them against excessive fatigue [35,67,68,69,70,71].

The current study also identified the main predictor of the impact of fatigue on health which is associated with lifestyle. The variable physical activity during the COVID-19 pandemic was indicated as the main factor determining the impact of fatigue on health in general and physical terms, while the number of meals consumed was the main factor determining the impact of fatigue on health in the cognitive and psychosocial terms.

Limitations and implications for the professional practice.

In the sample selection process, the authors of the study defined the study population, adopting the specificity of the education process in the nursing degree program as the main criterion, which includes both the theoretical and practical education components. The nursing student’s first contact with the practical dimension of the profession takes place while participating in clinical classes as early as the first year of the study, and the implementation of the practical professional training activities during the COVID-19 pandemic required additional effort from students and academic staff due to the epidemiological restrictions put in place.

The presented results of the authors’ own study provide new information on the factors determining the impact of fatigue on the health of Polish nursing students during the COVID-19 pandemic. Considering the negative consequences of the impact of fatigue on nursing students’ daily functioning and health, it is reasonable to conduct screening studies assessing the nursing students’ health status in all its dimensions. Moreover, it is necessary to introduce effective measures aimed at modifying all the proven fatigue predictors associated with the lifestyle. These measures may prove effective not only in combating fatigue but also in restoring the students’ optimal health. It is therefore important to implement institutional support for students as well as psychological assistance as required, especially in situations where, e.g., inappropriate dietary behaviors, are related to mental and physical health, or failure to perform physical activity results from experiencing excessive fatigue. The presented study is one of the first such large multi-center studies conducted during the COVID-19 pandemic at Polish universities. Its strengths include the large sample size as well as the inclusion of nursing students from different regions of Poland, who are a representative sample of the country. The study, however, has its limitations, especially those related to the lack of data on the levels of fatigue experienced by the nursing students and the predictors of fatigue (i.e., the number of meals consumed, the number of hours spent working on the computer, and the reduction in physical activity) before the pandemic period, which could be of importance to the current levels of the variables under analysis. Another limitation is the uneven distribution of the study sample in terms of gender. Since the nursing profession is dominated by women, the obtained results should not be generalized to the male population. Despite its limitations, this study provides important findings and can provide a starting point for broader research into the issues concerned.

## 5. Conclusions

Reduced physical activity in the study group of nursing students is the main predictor of the impact of fatigue on health in general and physical terms, while the number of meals consumed is the main predictor in cognitive and psychosocial terms.

The highest level of the impact of fatigue was noted for individuals who declared having only one or two meals a day. If students eat rationally and consume an appropriate number of meals, the impact of fatigue on health diminishes.

It was proven that fatigue affected the physical health of students depending on their age and the year of study. Students from the youngest age group (≤20) and first-year nursing students experienced a significantly greater impact of fatigue on health in general terms than in physical and cognitive terms. With an increase in the number of hours worked remotely, the level of fatigue related to psychosocial functioning increased as well.

It is recognized that action is needed to reduce the impact of fatigue on students’ health by modifying the predictors related to their lifestyles.

## Figures and Tables

**Figure 1 jcm-11-06034-f001:**
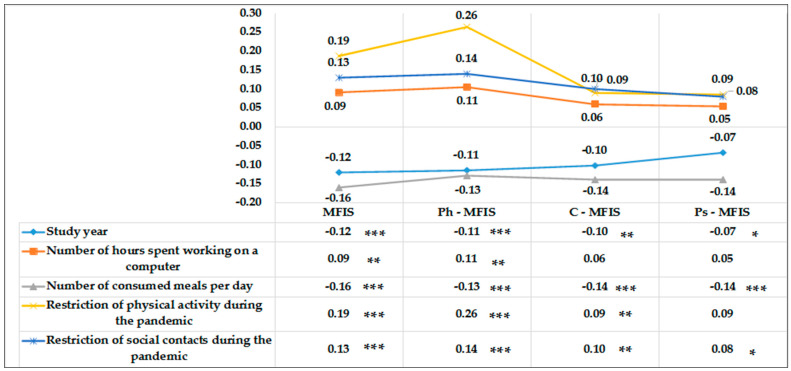
Correlation relationships between both the selected socio-demographic and lifestyle factors and the impact of fatigue on health in general, physical, cognitive, and psychosocial terms. Statistically significant: * *p* < 0.05; ** *p* < 0.00; *** *p* < 0.001.

**Table 1 jcm-11-06034-t001:** Study group characteristics.

Variables		Total N = 894	
		Number	%
Gender	female	822	91.95
	male	72	8.05
Study year	first	397	44.41
	second	289	32.33
	third	208	23.27
Age (years) M = 20.73; SD = 1.81	≤20	481	53.80
	21–22	319	35.68
	≥23	94	10.51
Place and form of residence	with family/someone close	621	69.46
	on their own	273	30.54
Number of hours spent working on a computer M = 6.08; SD = 3.19	≤5	433	48.43
	6–9	302	33.78
	≥10	159	17.79
Number of consumed meals per day M = 3.48; SD = 0.87	1–2	104	11.63
	3	382	42.73
	4	280	31.32
	≥5	128	14.32
Restriction of physical activity during the pandemic	no	211	23.60
	yes, to a small extent	161	18.01
	yes, to a medium extent	278	31.10
	yes, to a considerable extent	244	27.29
Subjective health status assessment during the pandemic	bad	24	2.68
	good/ average	613	68.57
	very good	257	28.75
Restriction of social contacts during the pandemic	very high	141	15.77
	considerable	360	40.27
	medium/ average	229	25.62
	to a small extent	164	18.34

Explanations: N—number of subjects.

**Table 2 jcm-11-06034-t002:** Descriptive statistics of the analyzed variables.

Variables		N = 894				
		M	95% CI	Me	Min.–Max.	SD
MFIS		57.67	56.68–58.65	57	21–102	15.02
MFIS (subscales)	Ph-MFIS	24.55	24.09–25.01	25	9–44	7.02
	C-MFIS	27.79	27.22–28.37	27	10–50	8.70
	Ps-MFIS	5.32	5.19–5.46	5	2–1	2.09

Explanation: N—sample size, M—arithmetic mean, 95% CI—confidence interval for the mean value, Me—median, Min.—minimum, Max.—maximum, SD—standard deviation, MFIS—Modified Fatigue Impact Scale, Ph-MFIS—fatigue impact on health in physical terms, C-MFIS—fatigue impact on health in cognitive terms, Ps-MFIS—fatigue impact on health in psychosocial terms.

**Table 3 jcm-11-06034-t003:** The results of the test for significance of the effect of socio-demographic and lifestyle variables on fatigue in general, physical, cognitive, and psychosocial terms.

Variables			Ph-MFIS					C-MFIS					Ps-MFIS				
			M	SD	F	*p*	NIR	M	SD	F	*p*	NIR	M	SD	F	*p*	NIR
Study year	first	A	25.58	6.84	7.76	***	A > B, C ***	28.78	8.66	4.94	**	A > B *, A > C **	5.52	2.11	3.34	*	A > B **
	second	B	23.71	7.08				27.26	8.69				5.13	2.07			
	third	C	23.76	7.04				26.65	8.62				5.22	2.04			
Age (years)	≤20	A	25.17	6.71	4.25	**	A > B *, A > C **	28.62	8.49	5.18	**	A > B *, A > C **	5.41	2.10	0.95	0.39	-
	21–22	B	23.92	7.42				27.06	8.84				5.21	2.10			
	≥23	C	23.51	6.90				26.07	8.84				5.26	1.98			
Number of hours spent working on a computer	≤5	A	23.82	7.04	4.58	**	A > B **	27.21	8.56	1.91	0.15	-	5.14	2.03	3.76	**	A > B **
	6–9	B	25.31	7.04				28.43	8.61				5.56	2.11			
	≥10	C	25.09	6.75				28.17	9.18				5.38	2.16			
Number of consumed meals per day	1–2	A	27.04	7.52	6.24	***	A > B, C, D ***	32.33	9.69	11.29	***	A > B, C, D ***	6.42	2.33	11.53	***	A > B, C, D ***
	3	B	24.69	6.91				27.47	8.50				5.24	2.02			
	4	C	23.93	6.37				27.02	7.99				5.13	2.00			
	≥5	D	23.48	7.79				26.78	8.91				5.10	2.02			
Restriction of physical activity during the pandemic	no	A	21.90	7.15	23.83	***	A < B **, A < C, D ***, B < D ***, C < D ***	26.86	8.71	4.04	**	A < D **, C < D **	5.27	2.32	7.31	***	A < D **, B < D ***, C < D ***
	yes, to a small extent	B	24.04	6.80				27.76	8.71				5.14	1.91			
	yes, to a medium extent	C	24.51	6.49				27.14	7.80				5.03	1.84			
	yes, to a considerable extent	D	27.22	6.69				29.37	9.45				5.83	2.17			
Restriction of social contacts during the pandemic	Restriction of social contacts during the pandemic	A	25.57	7.40	6.42	***	A < C *, A < D ***, B < C *, B < D **	29.55	9.50	3.36	*	A < C **, A < C **	5.65	2.33	2.44	0.06	-
	Restriction of social contacts during the pandemic	B	25.30	6.76				28.04	8.46				5.40	1.98			
	Restriction of social contacts during the pandemic	C	24.01	6.63				27.10	7.89				5.09	1.85			
	Restriction of social contacts during the pandemic	D	22.78	7.39				26.71	9.35				5.21	2.36			

Explanation: M—arithmetic mean; SD—standard deviation; F—Fischer’s test; MFIS—Modified Fatigue Impact Scale; Ph-MFIS—fatigue impact on health in physical terms; C-MFIS—fatigue impact on health in cognitive terms; Ps-MFIS—fatigue impact on health in psychosocial terms. Statistically significant: * *p* < 0.05; ** *p* < 0.01; *** *p* < 0.001.

**Table 4 jcm-11-06034-t004:** The summary of regression-predictors of the impact of fatigue on health in general, physical, cognitive, and psychosocial terms.

Variables		R^2^	*ßeta*	ß	t	*p*-Value
MFIS	Constant value			66.59	20.76	***
	Restriction of physical activity during the pandemic	0.04	0.15	2.04	4.55	***
	Number of consumed meals per day	0.06	−0.14	−2.47	−4.46	***
	Study year	0.07	−0.10	−1.91	−3.09	***
	Restriction of social contacts during the pandemic	0.08	−0.09	−1.44	−2.74	***
	R = 0.28; R^2^ = 0.08; corrected R^2^ = 0.08					
Ph-MFIS	Constant value			25.89	17.50	***
	Restriction of physical activity during the pandemic	0.07	0.23	1.47	7.08	***
	Number of consumed meals per day	0.09	−0.11	−0.89	−3.48	***
	Study year	0.09	−0.09	−0.83	−2.90	***
	Restriction of social contacts during the pandemic	0.10	−0.07	−0.54	−2.24	*
	R = 0.32; R^2^ = 0.10; corrected R^2^ = 0.10					
C-MFIS	Constant value			34.71	19.74	***
	Number of consumed meals per day	0.02	−0.13	−1.27	−3.88	***
	Restriction of social contacts during the pandemic	0.03	−0.09	−0.80	−2.60	***
	Study year	0.04	−0.09	−0.99	−2.74	***
	R = 0.20; R^2^ = 0.04; corrected R^2^ = 0.04					
Ps-MFIS	Constant value			6.68	15.75	***
	Number of consumed meals per day	0.02	−0.13	−0.31	−3.95	***
	Restriction of physical activity during the pandemic	0.03	0.07	0.12	1.89	*
	R = 0.18; R^2^ = 0.03; corrected R^2^ = 0.03					

Explanation: MFIS-Modified Fatigue Impact Scale, Ph-MFIS—fatigue impact on health in physical terms, C-MFIS—fatigue impact on health in cognitive terms, Ps-MFIS—fatigue impact on health in psychosocial terms. Statistically significant: * *p* < 0.05; *** *p* < 0.001.

## Data Availability

The data presented in this study are available on request from the first author.

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
