# Peer review of "Determinants of the Impact of Fatigue on the Health of Polish Nursing Students during the COVID-19 Pandemic"

_jcm, 2022, doi:10.3390/jcm11206034_

Round 1
Reviewer 1 Report
This is an interesting study that I like reading. The manuscript need some revisions. I report my comments/suggestions below:
- The introduction is clear and interesting but the Authors in their manuscript focused on a specific “group of young” that is nursing university students so I think that the Authors should insert some specific previous studies on the impact of covid-19 on university students and explore if there are some specific on nursing students. I can suggest the recent Cerutti et al.2022 published on Psychology in the Schools also to see the reference list.
- I suggest to clearly report the study’s hypothesis, what results did the authors expect to find?
- Why did the Authors decide to only examine nursing students? I think this information should be reported.
- 975 students agreed to participate and 894 were included that is the 91.69% of the total, but I think the Authors should report 975 on how many possible students have agreed to participate to clearly report the acceptance rate.
- The section participants should be more than a table
- I suggest to report the chronbach’s alpha values for the present study
- Statistical analysis, performing the one-way ANOVA did not allow to talk of effect of independent variables on dependent ones but this test allows to talk of differences between groups on several variables, thus I think the authors should be more cautious and modify the sentence.
- Paragraph 3.1 and 3.2 the authors reported that “Detailed data are provided in Table 2” but I think is table 3 and not 2.
- The regression models are my major concerns since the R2 values are too low to be considered interesting and significant in their results, also under the 10%. Furthermore, the Authors did not report Adjusted R2 values that usually are reported. In general, the models explained a too limited percentage of the variance in the dependent variables, so I suggest to remove the section related to the regression models.
- Check for typos and English
Author Response
29.09.2022
Dear Editors,
Thank you very much for a thorough editorial assessment of my manuscript, positive opinions, as well as the reviewers’ remarks. I used them as an important guide to improving the quality of my paper. The implemented corrections were done strictly according to their comments. All changes made in the text are marked in red. I have enclosed the re-edited manuscript and cover letter as responses to Reviewers, detailing how I followed their suggestions.
Thank you very much for your kind consideration of my paper.
Sincerely,
Kamila Rachubińska
Open Review
( ) I would not like to sign my review report
(x) I would like to sign my review report
English language and style
( ) Extensive editing of English language and style required
( ) Moderate English changes required
(x) English language and style are fine/minor spell check required
( ) I don't feel qualified to judge about the English language and style
Yes |
Can be improved |
Must be improved |
Not applicable |
|
Does the introduction provide sufficient background and include all relevant references? |
() |
(x) |
( ) |
( ) |
Are all the cited references relevant to the research? |
(x ) |
( ) |
( ) |
( ) |
Is the research design appropriate? |
(x) |
( ) |
( ) |
( ) |
Are the methods adequately described? |
(x) |
() |
( ) |
( ) |
Are the results clearly presented? |
() |
(x) |
( ) |
( ) |
Are the conclusions supported by the results? |
() |
(x) |
( ) |
( ) |
Comments and Suggestions for Authors
This is an interesting study that I like reading. The manuscript need some revisions. I report my comments/suggestions below:
RESPONSE: Thank you for your suggestions. We have added the missing data. The modifications introduced into the manuscript have been highlighted in colour.
- The introduction is clear and interesting but the Authors in their manuscript focused on a specific “group of young” that is nursing university students so I think that the Authors should insert some specific previous studies on the impact of covid-19 on university students and explore if there are some specific on nursing students. I can suggest the recent Cerutti et al.2022 published on Psychology in the Schools also to see the reference list.
RESPONSE: The authors have made every effort to make the section “Introduction” more valuable in form. An attempt has been made to clarify the background of the problem under study and the justification of the study undertaken among nursing students.
- I suggest to clearly report the study’s hypothesis, what results did the authors expect to find?
RESPONSE: As suggested by the reviewer, the authors of the study have introduced two research hypotheses.
- Why did the Authors decide to only examine nursing students? I think this information should be reported.
RESPONSE: In the subsection “Limitations and implications for the professional practice”, the authors of the study completed the validity of the study carried out on the group of nursing students.
- 975 students agreed to participate and 894 were included that is the 91.69% of the total, but I think the Authors should report 975 on how many possible students have agreed to participate to clearly report the acceptance rate.
RESPONSE: Changes have been introduced to the manuscript.
- The section participants should be more than a table.
RESPONSE: Subsection 2.2 has been complemented.
- I suggest to report the chronbach’s alpha values for the present study
RESPONSE: The information has been supplemented.
- Statistical analysis, performing the one-way ANOVA did not allow to talk of effect of independent variables on dependent ones but this test allows to talk of differences between groups on several variables, thus I think the authors should be more cautious and modify the sentence.
RESPONSE: Changes have been introduced to the manuscript.
- Paragraph 3.1 and 3.2 the authors reported that “Detailed data are provided in Table 2” but I think is table 3 and not 2.
RESPONSE: Changes have been introduced to the manuscript.
- The regression models are my major concerns since the R2 values are too low to be considered interesting and significant in their results, also under the 10%. Furthermore, the Authors did not report Adjusted R2 values that usually are reported. In general, the models explained a too limited percentage of the variance in the dependent variables, so I suggest to remove the section related to the regression models.
RESPONSE: The authors of the study ask that subsection 3.5 be left untouched, as they have conducted a literature review and concluded that there are many published study results in which the R² value is less than 10%. Together with experienced employees of the Faculty of Mathematics and Computer Science at the University of Warmia and Mazury in Olsztyn, who had been dealing with biostatistics for many years, the authors have verified both the results obtained and the correctness of the analyses conducted. Therefore, we consider it reasonable to leave section 3.5 in place.
- Check for typos and English
RESPONSE: As suggested by the Reviewer, final linguistic corrections were carried out by the translation office - OSCAR - Foreign Language School and Translation Office in Olsztyn.
Yours faithfully,
Kamila Rachubińska
Kamila Rachubińska, corresponding author
Department of Nursing, Faculty of Health Sciences, Pomeranian Medical University in Szczecin,
Head: Prof. Elżbieta Grochans
48 Żołnierska St., 71 – 210 Szczecin, Poland
Tel. (091) 48-00-910
E-mail: [email protected]

Reviewer 2 Report
I very much applaud the work you’re doing. But as best I can tell, the study has a fatal design flaw which largely invalidates the conclusions. Perhaps I am missing something, and am open to reviewing again if you can convince me of this.
Basically, I get the impression that the questionnaires were filled out one time. Presumably after an undefined length of lockdown. So the probability is that fatigue/depression more likely caused the lack of exercise and appetite rather than the other way around. So to me, I am not getting much useful information out of this data set. And I see no way to correct this. Even though you do give mention of this issue during the limitations, I do not see this as being a mild to moderate limitation, but rather one that makes the conclusions neither meaningful or probable
In submitting the manuscript elsewhere, you may want to clarify:
1- When during lockdown was the initial questionnaire filled in? How long approximately from the time lockdown was declared to the date the person filled out the questionnaire could be used to release approximate this
2- clarify whether there is any data collected on activity, diet etc. before onset of symptoms , which would allow it to be predictive of these variables actually causing the “fatigue,” or whether it was measuring activity, diet, etc. only after the onset of the fatigue. In which case the fatigue/depression may have caused the increased food intake and activity, instead of the factors causing the fatigue.
So clarify how soon after lockdown the questionnaires were filled out, and whether fatigue or these lifestyle changes came first.
Author Response
29.09.2022
Dear Editors,
Thank you very much for a thorough editorial assessment of my manuscript, positive opinions, as well as the reviewers’ remarks. I used them as an important guide to improving the quality of my paper. The implemented corrections were done strictly according to their comments. All changes made in the text are marked in red. I have enclosed the re-edited manuscript and cover letter as responses to Reviewers, detailing how I followed their suggestions.
Thank you very much for your kind consideration of my paper.
Sincerely,
Kamila Rachubińska
Open Review
(X) I would not like to sign my review report
( ) I would like to sign my review report
English language and style
( ) Extensive editing of English language and style required
( ) Moderate English changes required
(x) English language and style are fine/minor spell check required
( ) I don't feel qualified to judge about the English language and style
Yes |
Can be improved |
Must be improved |
Not applicable |
|
Does the introduction provide sufficient background and include all relevant references? |
() |
(x) |
( ) |
( ) |
Are all the cited references relevant to the research? |
(x ) |
( ) |
( ) |
( ) |
Is the research design appropriate? |
( ) |
( ) |
(x) |
( ) |
Are the methods adequately described? |
( ) |
(x) |
( ) |
( ) |
Are the results clearly presented? |
() |
(x) |
( ) |
( ) |
Are the conclusions supported by the results? |
() |
( ) |
(x) |
( ) |
Comments and Suggestions for Authors
I very much applaud the work you’re doing. But as best I can tell, the study has a fatal design flaw which largely invalidates the conclusions. Perhaps I am missing something, and am open to reviewing again if you can convince me of this.
RESPONSE: Thank you for your suggestions. We have added the missing data.
The modifications introduced into the manuscript have been highlighted in colour.
Basically, I get the impression that the questionnaires were filled out one time. Presumably after an undefined length of lockdown. So the probability is that fatigue/depression more likely caused the lack of exercise and appetite rather than the other way around. So to me, I am not getting much useful information out of this data set. And I see no way to correct this. Even though you do give mention of this issue during the limitations, I do not see this as being a mild to moderate limitation, but rather one that makes the conclusions neither meaningful or probable
RESPONSE: The authors of the study have attempted to clarify the background of the investigated problem and the justification of the study undertaken among nursing students and have introduced two research hypotheses. In the subsection “Limitations and implications for the professional practice”, the authors indicated the validity of the study carried out on the group of nursing students. The authors confirm that the study has its limitations, especially those related to the lack of data on the levels of fatigue experienced by the nursing students and the predictors of fatigue before the pandemic period, which may be of importance to the current levels of the variables under analysis. Despite its limitations, the study concerned yields important findings and may provide a starting point for broader research into the issues in question.
The authors would like to thank the reviewers for the invaluable comments and guidelines which will be used for further research in this field.
In submitting the manuscript elsewhere, you may want to clarify:
1- When during lockdown was the initial questionnaire filled in? How long approximately from the time lockdown was declared to the date the person filled out the questionnaire could be used to release approximate this
RESPONSE: Subsection 2.1 “Settings and design” indicates that the study was conducted between 20 March 2021 and 15 December 2021, in the second year of the Covid-19 pandemic.
2- clarify whether there is any data collected on activity, diet etc. before onset of symptoms , which would allow it to be predictive of these variables actually causing the “fatigue,” or whether it was measuring activity, diet, etc. only after the onset of the fatigue. In which case the fatigue/depression may have caused the increased food intake and activity, instead of the factors causing the fatigue.
RESPONSE: We would like to clarify that we have no data on the physical activity, diet, etc., of nursing students prior to the onset of the fatigue symptoms. Despite this limitation, the study concerned yields important findings and may provide a starting point for broader research into the issues in question. We are grateful for the invaluable comment, and we will use this guideline for further research in this field.
So clarify how soon after lockdown the questionnaires were filled out, and whether fatigue or these lifestyle changes came first.
RESPONSE: The study was conducted between 20 March 2021 and 15 December 2021, in the second year of the Covid-19 pandemic. We would like to clarify that fatigue referred to as “lockdown fatigue”, is defined as a state of energy loss and exhaustion resulting from the restrictions imposed due to the pandemic, affecting all aspects of the biopsychosocial functioning of humans. An important cause of lockdown-induced fatigue, identified by many researchers, is the necessity to significantly modify the existing routine activities, not only in the professional or educational sphere but also in the implementation of health-promoting behaviours such as physical activity, dietary habits, or sleep. The deterioration of health in physical, mental, and behavioural terms appears to be the proven and most important consequence of pandemic-induced fatigue. It is, therefore, reasonable to regard the fatigue induced due to the destructive effect of the Covid-19 pandemic on the functioning of young people as a serious problem. One of the ways to reduce the adverse fatigue impact on health is to take actions aimed at modifying its predictors, especially those related to the lifestyle, hence the attempt to identify them in this study.
Yours faithfully,
Kamila Rachubińska
Kamila Rachubińska, corresponding author
Department of Nursing, Faculty of Health Sciences, Pomeranian Medical University in Szczecin,
Head: Prof. Elżbieta Grochans
48 Żołnierska St., 71 – 210 Szczecin, Poland
Tel. (091) 48-00-910
E-mail: [email protected]

Round 2
Reviewer 2 Report
Accepted for publication